# A Novel Air Quality Monitoring Unit Using Cloudino and FIWARE Technologies

**Yolanda Raquel Baca Gómez [1,\*], Hugo Estrada Esquivel [1,\*], Alicia Martínez Rebollar [2,\*]**  **and Daniel Villanueva Vásquez [1,3,\*]**

1    National Council of Science and Technology (CONACyT), 03940 Mexico City, Mexico
2    National Center for Research and Technological Development (CENIDET), 62490 Cuernavaca, Mexico
3    Center for Research and Innovation in Information and Communication Technologies (INFOTEC), 14050 Mexico City, Mexico
\*    Correspondence: yolanda.baca@cns-ipicyt.mx (Y.R.B.G.); hugo.estrada@conacyt.mx (H.E.E.); amartinez@cenidet.edu.mx (A.M.R.); daniel.villanueva@infotec.mx (D.V.V.)

**Abstract:** Smart City applications aim to improve the quality of life of citizens. Applying technologies of the Internet of Things (IoT) to urban environments is considered as a key of the development of smart cities. In this context, air pollution is one of the most important factors affecting the quality of life and the health of the increasing urban population of industrial societies. For this reason, it is essential to develop applications that allow citizens monitoring the concentration of pollutants and avoid places with high levels of pollution. Due to the increasing use of IoT in different areas, there are arising platforms which deal with the challenges IoT implies, such as FIWARE, which provides technologies to facilitate the development of IoT applications. In this paper, an Air Quality Monitoring Unit using Cloudino and Arduino devices and FIWARE technologies is presented. Through Cloudino and Arduino, the monitoring unit gather data from various sensors and transforms the data in a FIWARE data model. Then, the measurements are sent to the Orion Context Broker (OCB), which is a software component provided by FIWARE. The Orion Context Broker allows to manage and publish the data to be consumed by users and applications.

**Keywords:** smart cities; air quality; IoT; FIWARE; data model; Cloudino

## 1. Introduction

Today, citizens have huge concerns about the quality of life in their cities, especially regarding the level of pollution. Air quality level is of great importance not only in planning our activities, but also in taking precautionary measures for our health [1]. Air pollution is one of the most important factors affecting the quality of life and the health of the increasing urban population of industrial societies [2].

Therefore, several proposals emerged to attack the problem of pollution and the monitoring of air quality [3–8]. Some of these proposals are framed in the integration between Information and Communication Technologies (ICT) and Internet of Things (IoT) [9]. The latter is designed to support the Smart City vision, which aims at exploiting the most advanced communication technologies to support added-value services for the administration of the city and for the citizens, which are the primary beneficiaries of Smart City initiatives, either directly or indirectly [10,11].

IoT is a new revolution of Internet, providing interaction among the real (physical) and virtual (digital) worlds. It makes objects themselves recognizable, obtains intelligence, and communicates information that has been aggregated by other things, allowing people and things to be connected anytime and anyplace with anything and anyone [12]. It also introduces new opportunities, such as the capability to monitor and manage devices remotely and to analyze and take actions based on the

information received from various real-time traffic data streams with the aim of improving the quality of life of citizens and the management of resources [13,14]. Due to the growing awareness of IoT, IoT platforms were raised, such as FIWARE (https://www.fiware.org/), which is a standard platform for developing Smart City applications. It was launched by the European Commission and aims to develop the core future technologies in the IoT paradigm.

In this paper, we present a novel air quality monitoring unit, which uses Cloudino (http://www.cloudino.io/) and Arduino devices. The goal of this monitoring unit is to measure the air quality. Also, our proposal has been implemented on FIWARE platform. The main contributions of this proposal include our monitoring unit, which is linked to an application called Green Route that allows users to determine the best route to reach a destination, taking into account the air pollution. The data is represented in a data model, allowing interoperability between the applications that use it. Additionally, the data is available in the FIWARE platform and any user or application can consume the data.

The air quality monitoring unit measures the $CO_2$, CO, VOC, HC, $NH_3$, and NOx pollutants. In addition, it measures the altitude, barometric pressure, relative humidity, temperature, and with a GPS, it obtains the geolocalization. Our proposal was implemented and is being used in Mexico City, one of the most polluted cities in the world. Furthermore, our application was implemented using the technology and components of the FIWARE platform. Thus, the combination of Cloudino and FIWARE provides a monitoring unit able to be remotely configured and automatically connected to the FIWARE Cloud to send the data of the sensors to the Orion Context Broker. Once the data is in the Orion Context Broker, it can be queried by the applications of the FIWARE platform users or by any other user.

## 2. Related Work

The monitoring of urban air quality through mobile sensing is a popular topic, especially with the possible applications of designing, deploying and administrating sensors that measure gaseous pollutants from urban scenarios [3]. Continuously monitoring air quality and making the real-time data available to public is important and useful to the residents, particularly those living in big urban areas [4]. Currently, there are several works related to air quality monitoring through the measurement of some pollutants in the air.

The low-cost Air Quality System presented in [3] proposes a concept of an air quality monitoring system through mobile sensing using low-cost reliable sensors in order to record the variation of the CO in metropolitan areas. The system architecture consists of portable personal sensing devices that would normally be attached to the car of a driver or volunteer as they drive. The personal sensing devices communicate with an application on the driver's mobile phone that gathers the measured data from the sensing device, then tags it with additional information regarding the GPS coordinates of the location where that data was gathered, along with the time and date. A cloud-based server receives and stores large amounts of information from multiple users as they finish pollution recording sessions and upload the gathered data over the Internet. The server will then parse and interpret the data, applying interpolation models in order to generate a pollution level heat-map and make that map available online on the project's website.

The work presented in [4] proposes a smart sensor system for air quality monitoring which consists of three units: The smart sensor unit, the smartphone, and a server. The smart sensor unit utilizes sensors to sample air quality data anytime and anywhere, and sends the data to the smartphone through Bluetooth. The smartphone displays the data to the user and relays the data to the server for processing, visualization, air quality monitoring, and web publishing. The air quality prototype system is capable of measuring $PM_{2.5}$, CO, $CO_2$, VOC, temperature, and humidity, and detecting other hazard gases. Air quality data from individual submission at different locations, different dates, and different times are collected, stored, and processed in real-time at the server side in cloud. The data can

be visualized on the map in terms of the location, date, and time that the data were collected. Color patterns are used to visualize the air quality.

The device built in [5] can acquire information about the air quality of its surroundings, store it in a temporary memory buffer, and periodically relay it to a central online repository. The system is composed by two major components: A mobile unit and an online web service. The mobile unit gathers the measurements of the sensors, which are transmitted by GSM link to the web server and displayed live on a liquid crystal display. The online web server provides user access to pollution statistics. Real-time gathered data can be freely accessed by the public through an online web interface. Users can select and view different gasses and concentrations overlapped on a map of the city. The device measures the following compounds: CO, NOx, and HC.

The air pollution monitoring system proposed in [6] consists of a transmitter and receiver part. The transmitter part is integrated by a single-chip microcontroller, an air pollution sensors array, a General Packet Radio Service Modem (GPRSModem), and a Global Positioning System Module (GPS-Module) for transmitting the information. The receiver part is a pollution server with Internet connectivity and a database unit connected to various wired and wireless clients for further retrieval information. The transmitter part consists of a Mobile Data-Acquisition Unit (Mobile-DAQ) and a fixed Internet-Enabled Pollution Monitoring Server (Pollution-Server). The Mobile-DAQ unit gathers air pollutants levels (CO, $NO_2$, and $SO_2$), and packs them in a frame with the GPS physical location, time, and date. The frame is subsequently uploaded to the GPRS Modem and transmitted to the Pollution-Server via the public mobile network. A database server is attached to the Pollution-Server for storing the pollutants level for further usage by various clients.

The vehicular-based mobile approach for measuring fine-grained air quality in real-time proposed in [7] considers two cost-effective data farming models: The first can be deployed on a public transportation, and the second on a personal sensing device. Both are focused only on CO. A cloud service to gather data was implemented along with a web portal, which allows users to view real-time pollution data. The system uses Arduino to control and communicate with the carbon monoxide sensor, dust sensor, GPS, and the cellular modem. The readings are sampled periodically and transmitted to the cloud server.

The approach followed in [8] in the UrVAMM project proposes a solution to acquire air quality information at street level and help reduce vehicle emissions. UrVAMM unites two existing systems, respectively owned by the two partners in the project, Ingenieros Asesores (IA) and ADN Mobile Solutions (ADN). IA's NanoEnvi is an experimental portable system for the measurement of the most relevant atmospheric pollutants in urban areas (CO and NOx) on the road. ADN's CatedBox is a technology and a methodology that collaborates to enable continuous education of drivers towards efficient driving in terms of lower fuel consumption rates and pollutant emissions. As previously stated, this work intends to highlight the collaboration between both systems, but emphasizes the novelties of NanoEnvi in the topic of mobile air quality analysis.

To conclude, in the related work, data models are not used, and the way the data is shared with users is not specified. Data models are necessary to ensure the interoperability between different applications and to guarantee the access to the data by any kind of user. In this work, technologies to facilitate sharing and access the data are presented. Moreover, almost all the pollutants measured in this work match those measured in the related work. In this research, a few more pollutants were included. For statistical reasons, it is essential to publish and share as much information as possible about air quality, and to define mechanisms that allow to most of the people to access this kind of data. In our proposal, the data can be accessed by users in the Green Route Application, but can also be consumed by any kind of application or users through REST calls to the Orion Context Broker.

## 3. Background About Air Quality Pollutants

Air pollution is a mixture of solid particles and gases in the air, including car emissions, chemicals from factories, dust, pollen, and mold spores that may be suspended as particles. Air pollution may

cause problems to our health, such as asthma, cough, and lung disorders. In addition, the pollution can cause global warning, acid rain, and disturbing plant growth [15].

The most common pollutants take into account for the Air Quality Index are $CO_2$, $NO_2$, $O_3$, PM 2.5, PM 10, and $SO_2$ [1]. The air quality pollutants considered in this research work are the following: VOCs, CO, $CO_2$, NOx, HC, $NH_3$, and $N_2S$. However, it is easy to add new sensors to the Air Quality Monitoring Unit and gathering other kind of pollutants. The pollutants measured by the Air Quality Monitoring Unit and its effects on human health are briefly described below:

- *Volatile Organic Compounds (VOCs):* These are substantial contributors to the formation of ozone and other photochemical oxidants. They are largely used in industries as solvents or chemical intermediates and can present a risk for human health. Some VOCs have been identified as toxic or mutagenic pollutants at concentrations sometimes present in urban environments [16].
- *Carbon Monoxide (CO):* This is generated by incomplete combustion of carbon. Even relatively small amounts of it can lead hypoxic injury, neurological damage, and possibly death [5].
- *Carbon Dioxide ($CO_2$):* This is the main greenhouse gas that causes global warming and climate change. Due to human activity, the concentration of $CO_2$ in the atmosphere is increasing at a fast speed [4]. It can increase the likelihood of asthma attacks and may cause a rise in asthma cases among children [5].
- *Nitrogen Oxides ($NO_X$):* These are produced from the burning of fuel in the engine. Road traffic is responsible for 49% of all $NO_X$ emissions. $NO_X$ helps to provoke acid rain. They also combine with hydrocarbons to form low-level ozone pollution and may contribute to lung disease [5].
- *Hydrocarbons (HC):* These are compounds of hydrogen and carbon and are present in petrol and diesel. Benzene is an example of a hydrocarbon. Road traffic is responsible for about 35% of all HC emissions. Hydrocarbons are carcinogenic and a major ingredient of smog. They are often found in gasoline and diesel exhaust, which are major pollutants in populated areas. Exposure to this mixture may result in asthma attacks, increase likelihood of cancer, chronic exacerbation of asthma, and other health problems [5].
- *Ammonia ($NH_3$):* One of the most widespread gases, children with asthma may be particularly sensitive to ammonia fumes. A significant part of respiratory allergies are related to this gas and prolonged exposure to ammonia may cause nasopharyngeal and tracheal burns, bronchiolar and alveolar edema, and airway destruction, resulting in respiratory distress of failure [5].
- *Hydrogen Sulfide ($N_2S$):* This is generated by bacteria as part of organic material decomposition. It can cause asthma attacks, eye, throat, and lung irritation, nausea, headache, nasal blockage, sleeping difficulties, weight loss, and chest pain.

## 4. Air Quality Monitoring Unit Architecture

The Air Quality Monitoring Unit allows the measurement of some pollutants that are harmful to human health and other data, such as altitude, barometric pressure, relative humidity, and temperature, through different sensors. The gathered data is stored by using a data model in the cloud and can be consumed by any kind of user or application.

Our proposal of the Air Quality Monitoring Unit architecture is shown in the Figure 1, and is composed by sensors and electronic components, an acquisition and processing and cloud connection module based on Cloudino, and FIWARE Ecosystem components. The monitoring unit communicates data to the FIWARE Cloud using a specific data model to ensure the interoperability. All the gathered information by the different components is shown in an LCD Display, and is available in the FIWARE Cloud.

Finally, all data available in the FIWARE Cloud can be used by any kind of user or application. Each one of these components is explained in detail in following sections. The data published in the Orion Context Broker by the Air Quality Monitoring Unit is consumed by the Green Route Application.

The usage of a data model ensures the interoperability of the application, and when the information is published in the Orion Context Broker it can be consumed by any user or application.

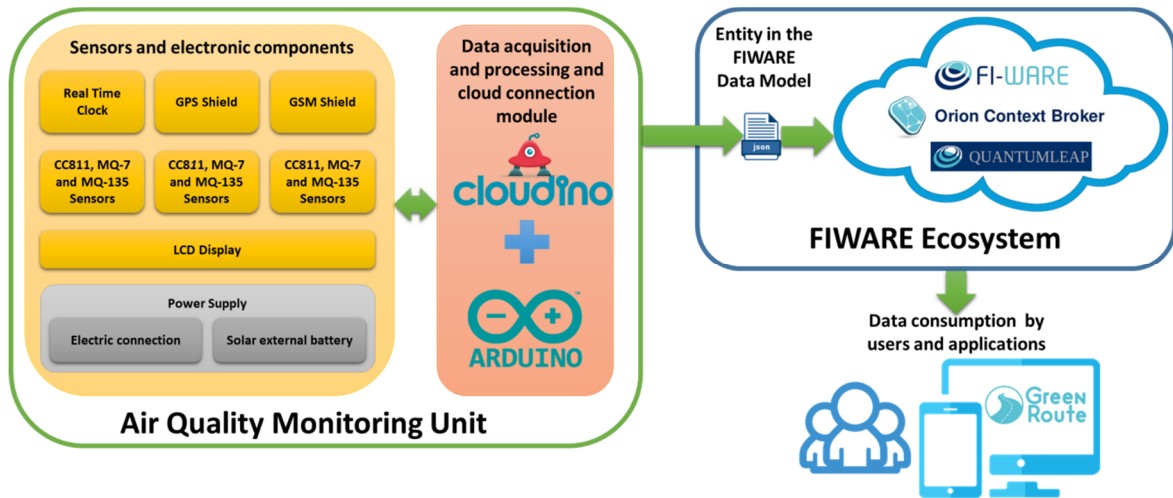

**Figure 1.** Air Quality Monitoring Unit architecture.

## 4.1. Sensors and Electronic Componentes

The sensors and electronic components that make up the Air Quality Monitoring Unit are the following:

- *Real Time Clock*: A clock that keeps track of the current time and can be used to program actions at a certain time or know at what time an action occurred in the Air Quality Monitoring Unit. It can record the time at which measurements were collected in order to identify at what time there is a higher concentration of pollutants.
- *GPS Shield:* The GPS Shield provides the location of the Air Quality Monitoring Unit.
- *GSM Shield*: The GSM Shield allows the Air Quality Monitoring Unit to connect to the Internet if there are no WiFi networks available.
- *CCS811 Sensor*: An ultra-low power digital gas sensor used to detect a wide range of (VOCs) and $CO_2$ (Carbon Dioxide).
- *MQ-7 Sensor*: Suitable for sensing CO concentrations in the air, and has a high sensitivity and fast response time.
- *MQ-135 Sensor*: Senses the gases like $NH_3$ (ammonia), $NO_X$ alcohols, aromatic compounds, $N_2S$ (Hydrogen Sulfide), benzene, steam, and other harmful gases.
- *GA1A1S202WP Sensor:* With the data obtained by the analog light sensor, the Air Quality Monitoring Unit can regulate the lights in order to save electricity.
- *BME280 Sensor:* This sensor allows to measure the barometric pressure, altitude, temperature, and relative humidity.
- *LCD Display*: This component shows the values of the current measurements.
- *Power Supply*: The Air Quality Monitoring Unit supports two types of power supply: y electric connection or by solar external battery. If there is no electric connection available, it can be powered by the battery.

## 4.2. Entity Definition Based in the AirQualityObserved FIWARE Data Model

FIWARE provides data models (https://github.com/Fiware/dataModels/blob/master/specs/howto.md) in order to facilitate the exchange of data between applications. These data models were defined in relation to the FIWARE reference context model: Open Mobile Alliance Next Generation

Service Interfaces (OMA-NGSI). This does not imply they cannot be used outside of the NGSI context model, but does indicate that some of the design principles have been driven by that.

The standard NGSI v2 of FIWARE is intended to manage the entire lifecycle of context information, including updates, queries, registrations, and subscriptions. The main elements in the NGSI data model are context entities, attributes, and metadata, as shown in Figure 2. The difference among these elements is that attributes describe the entity and metadata describe attributes [17,18]:

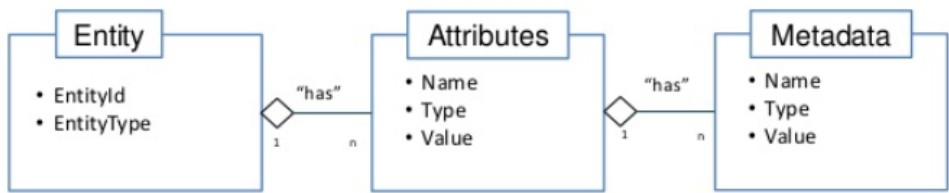

**Figure 2.** Elements in the NGSI data model.

Context entities, or simply entities, are the center of gravity in the FIWARE NGSI information model. An entity represents a thing, which is any physical or logical object (e.g., a sensor, a person, a room, an issue in a ticketing system, and so on). Each entity has an entity ID and entity type. Entity types are intended to describe the type of thing represented by the entity. Each entity is uniquely identified by the combination of its ID and type. Context attributes are properties of context entities. In the NGSI data model, attributes have an attribute name, an attribute type, an attribute value, and metadata.

For our proposal, we created an entity based on the AirQualityObserved (https://fiware-datamodels.readthedocs.io/en/latest/Environment/AirQualityObserved/doc/spec/index.html) FIWARE data model. The purpose of the entity is to send the Air Quality Monitoring Unit data to the Orion Context Broker. The AirQualityObserved data model allows us to generate an observation of air quality conditions at a certain place and time. In order to enable a proper management of the concentrations of the different pollutants, it is recommended that there is to be an attribute for each pollutant, which name must be exactly equal the chemical formula of the measurand.

In Figure 3, the entity of the Air Quality Monitoring unit is presented. The *id* field uniquely identifies the entity. The *type* field establishes the data model. In this case, the AirQualityObserved data model is used. The *address* field specifies the address in which the Air Quality Monitoring Unit is located. The *dateObserved* field shows the date and hour when the data was sent to the Orion Context Broker. The *location* field specifies the coordinates where the Air Quality Monitoring Unit is located. The *source* field establishes the data source. In this case, the data is obtained from the Air Quality Monitoring Unit prototype. Finally, the *altitude*, *barometricPressure*, *relativeHumidity*, *temperature*, *luminosity*, $CO_2$, *TVOC*, *CO*, *NOx*, $NH_3$, and $N_2S$ fields show the values of the sensors measurements.

*4.3. Data Acquisition and Processing and Cloud Connection Module*

The data acquisition and processing and cloud connection module was built using Cloudino (http://cloudino.io/) and Arduino. Cloudino is used as a microcontroller dedicated to the network layer, working in parallel with the Arduino (https://www.arduino.cc/) microcontroller to gather and process the data obtained from the sensors and electronic components. There are a variety of sensors and actuators that are compatible with Arduino. The main advantage over other platforms is the large amount of resources available, both in terms of software and hardware [19]. Cloudino is also responsible for the Internet connection through a WiFi network, or through GSM if there is no WiFi network available. Some Cloudino's functions were implemented in the Arduino code to carry out the data acquisition and process the data.

The functionalities of the Air Quality Monitoring Unit are controlled through the Arduino by using different functions provided by Cloudino and with the corresponding libraries of the electronic components. Cloudino provides a new graphical interface in which the Arduino can be programmed

in a remote way. In Figure 4, an example of the code for gathering data defined in Arduino through the Cloudino interface is presented.

```json
{
    "id":"AirQualityMonitoringUnit",
    "type":"AirQualityObserved",
    "address": {
        "type": "StructuredValue",
        "value": {
            "addressCountry": "MX",
            "addressLocality": "Ciudad de México",
            "streetAddress": "San Fernando"} },
    "dataSource":{
        "type":"text",
        "value":"Monitoring unit"},
    "dateObserved":{
        "type":"DateTime",
        "value":"2018-03-14T17:00:00-05:00"},
    "location": {
        "value": {
            "type": "Point",
            "coordinates": [-99.122984, 19.431768]},
        "type": "geo:json"},
    "altitude":{
        "type":"float",
        "value":2140.93},
    "barometricPressure":{
        "type":"float",
        "value":781.18},
    "relativeHumidity":{
        "type":"float",
        "value":29.54},
    "temperature":{
        "type":"float",
        "value":25.38},
    "luminosity":{
        "type":"float",
        "value":8.49},
    "CO2":{
        "type":"float",
        "value":2},
    "TVOC":{
        "type":"float",
        "value":4.5},
    "CO":{
        "type":"float",
        "value":1.1},
    "NOx":{
        "type":"float",
        "value":47},
    "NH3":{
        "type":"float",
        "value":1},
}
```

**Figure 3.** Air Quality Monitoring Unit entity.

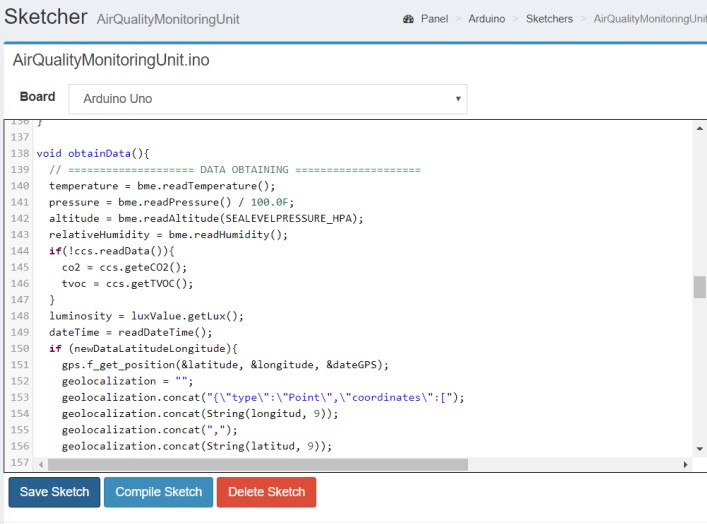

**Figure 4.** Example of Arduino's code through Cloudino's interface.

Furthermore, Cloudino provides a functionality to automatically transform the data into the Air Quality Monitoring Unit entity, which is based on the AirQualityObserved FIWARE data model and sends the data to the Orion Context Broker. In Figure 5, the Cloudino's interface, which allows us to link the Air Quality Monitoring Unit with the Orion Context Broker, is presented.

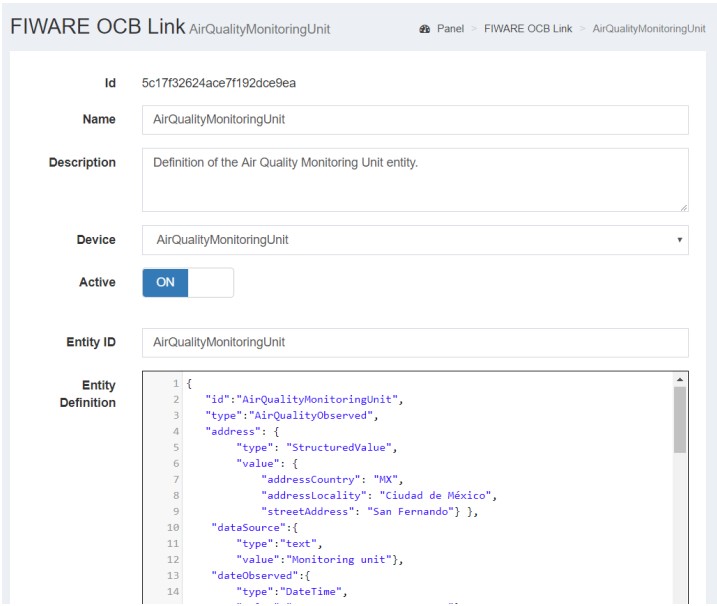

**Figure 5.** Example of the Cloudino's FIWARE Orion Context Broker Link.

Finally, as a result of the development of this project, a guide called: *Creating your air quality sensor with Cloudino* (https://guided-tour-smartsdk.readthedocs.io/en/latest/sensors/cloudino/) was published, which explains the steps to configure Cloudino and Arduino to send data to the Orion Context Broker.

*4.4. FIWARE Ecosystem*

Today, FIWARE is an open source API platform, and it has released several standardized software components aimed at helping startups and enterprises build the next generation of smart applications and services for cities, industries, e-health, or agribusiness. FIWARE provides a set of tools for different functionalities. It is an innovative ecosystem for the creation of new applications and Internet services. It is especially useful in terms of smart cities because it ensures the interoperability and the creation of standard data models.

The data is available in the Orion Context Broker and can be consumed by users and other applications. The data can be acquired through a subscription at the Orion Context Broker. The FIWARE components used in the Air Quality Monitoring Unit are as follows.

4.4.1. Orion Context Broker

The Orion Context Broker is an implementation of the NGSI REST API and is the key component of FIWARE enabling the data context ingestion and the subscription of applications to the data context. The main concept of the Context Broker is that data context producer can generate information and placed this in the cloud without a previous knowledge of the users or application that will use the data. The Orion Context Broker is able to handle context information at large scale by implementing standard REST APIs and allows developers to manage the whole lifecycle of context information, including updates, queries, registrations, and subscriptions [17,20–22].

The context information may come from many different sources. For example, Green Route is an application developed as a part of the SmartSDK FIWARE project.

The Orion Context broker is a FIWARE component commonly used to build applications that use data coming from the environment. All the information that is sent to this component needs to be represented in the NGSI format to allow sharing in the cloud and ensure the interoperability between different applications.

### 4.4.2. QuantumLeap

QuantumLeap (https://smartsdk.github.io/ngsi-timeseries-api/) is an implementation of an API that supports the storage of NGSI FIWARE NGSIv2 data into a time-series database known as ngsi-tsdb. The workflow of the QuantumLeap implementation consists on gathering information from any source, transform it into a FIWARE data model, and send it to Orion Context Broker. After, the context information stored in the Orion Context Broker notifies the QuantumLeap component through a subscription previously defined in the Orion Context Broker. Then, QuantumLeap persists the data as historic timeseries records in the CrateDB database. Finally, the information stored in the database is queried and visualized using Grafana (https://grafana.com/).

Grafana is a powerful tool with a query editor that allows us to choose among the metrics already registered and perform with them different visualizations. Grafana offers a web interface for the creation of tables, alerts, aggregation functions (sum, max, min, average), heat maps, and timeseries graphs. It also supports different databases and allows us to query them with their respective languages and sentences.

### 4.5. Data Consumption by Users and Applications

The data captured by the monitoring unit are available in the Orion Context Broker and can be consumed by users and other applications. The data can be acquired through a subscription to the Orion Context Broker. The subscription is a JSON to define the entity from which we want to obtain the information. In this case, it is the Air Quality Monitoring Unit entity. The data in the entity that must change in order to send the request and the URL of the service that is going to receive the information must be specified. As noted before, Green Route is one of the applications that uses the data of the monitoring unit with the purpose of improving mobility in cities.

The goal of Green Route is to help the user determine the best route to reach a destination in the city, taking into account their profile (for example, their health conditions), their preferences, such as the type of transport used, as well as information about weather, traffic, pollen, pollution, and alerts from other users. To achieve its goal, Green Route was developed as a platform for the consumption of multiple data sources, such as weather open data, environmental monitoring units, user sensors, web application information, and user generated information. All the information is concentrated in the FIWARE cloud and is used by Green Route for the generation of visualizations in city maps or historical data graphs of each type of data. Green Route will propose to the user the ideal route to avoid roads with high levels of pollution, traffic jams, pollen, and so on. It can be particularly useful for people with respiratory diseases or those who use alternative means of transport, such as bicycles.

The purpose of the Air Quality Monitoring Unit is to be a source of information to Green Route or other applications. Therefore, when the values of the sensor's measurements change in the Air Quality Monitoring Unit entity registered in the Orion Context Broker, the Green Route Application will get an asynchronous notification with the data gathered by the Air Quality Monitoring Unit through the subscription. This way, it is not necessary to continuously repeat query requests.

The elements of the subscription to the Air Quality Monitoring Unit Entity are shown in Figure 6. The *entities* field specifies the entity from which we need to receive notifications. The *condition* field contains the attributes that must change to send the notification. This could be one or more attributes. In this example, only one attribute is specified. The *notification* field specifies where the notification will be sent. In this case, the data is going to be sent to the Green Route Application. The *attrs* field specifies the values that will be sent. This could be one or more or all attributes of the entity. In this example, only three attributes are specified. The *expires* field specifies the date when the subscription

will stop working. Finally, the *throttling* field specifies the minimal period of time in seconds which must elapse between two consecutive notifications.

```
{
    "description": "Subscription to Monitoring Unit",
    "subject": {
        "entities": [
            {
                "id": "AirQualityMonitoringUnit",
                "type": "AirQualityObserved"
            }
        ],
        "condition": {
            "attrs": [
                "CO2"
            ]
        }
    },
    "notification": {
        "http": {
            "url": "https://greenroute.com:8080/notifications",
            "method": "POST"
        },
        "attrsFormat": "keyValues",
        "attrs": [
            "CO2",
            "TVOC",
            "temperature",
        ]
    },
    "expires": "2020-04-05T14:00:00.00Z",
    "throttling": 1
}
```

**Figure 6.** Subscription to the Air Quality Monitoring Unit entity.

## 5. Results and Discussion

The Air Quality Monitoring Unit provides the implementation of hardware and software for air pollution monitoring. Moreover, the data is available in the FIWARE Orion Context Broker, so that any application can query the measurements. For instance, in Figure 7, data obtained through the Orion Context Broker from different Air Quality Monitoring Units is presented in a graphic using Grafana (https://grafana.com/), an open platform for analytics and monitoring. Then, the historical information can be used to identify the correlation between pollutants, or the behavior of the pollutants in time lapses. For example, the air could be more contaminated at mornings when most of the people go to work or school.

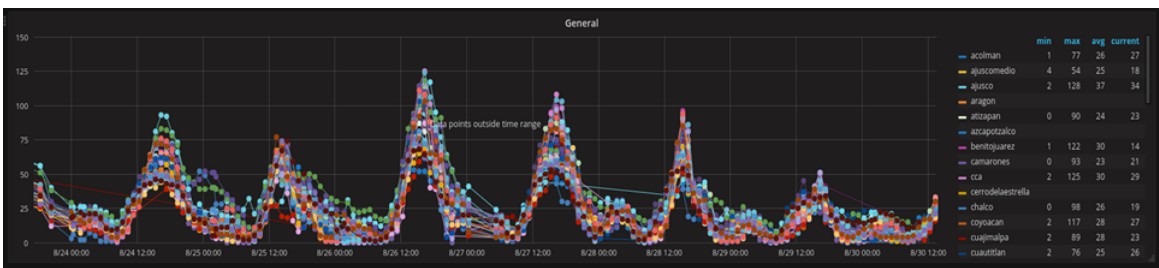

**Figure 7.** Air Quality Monitoring Unit data.

In addition, Cloudino allows a directly connection from the Air Quality Monitoring Unit to the FIWARE Orion Context Broker in the cloud in an easy way. One of the most important things is that the Cloudino allows the reprogramming of the Arduino via WiFi or Cloud. Therefore, is not necessary to be close to the Air Quality Monitoring Unit to reprogram it. In the Figure 8, an example of data collected by the Air Quality Monitoring Unit from Mexico City and sent to the Green Route Application [23,24] is shown. Finally, a generic data model is used, making it easier to share and publish the data. Anyone can view the data published in the Orion Context Broker and use it in their own applications.

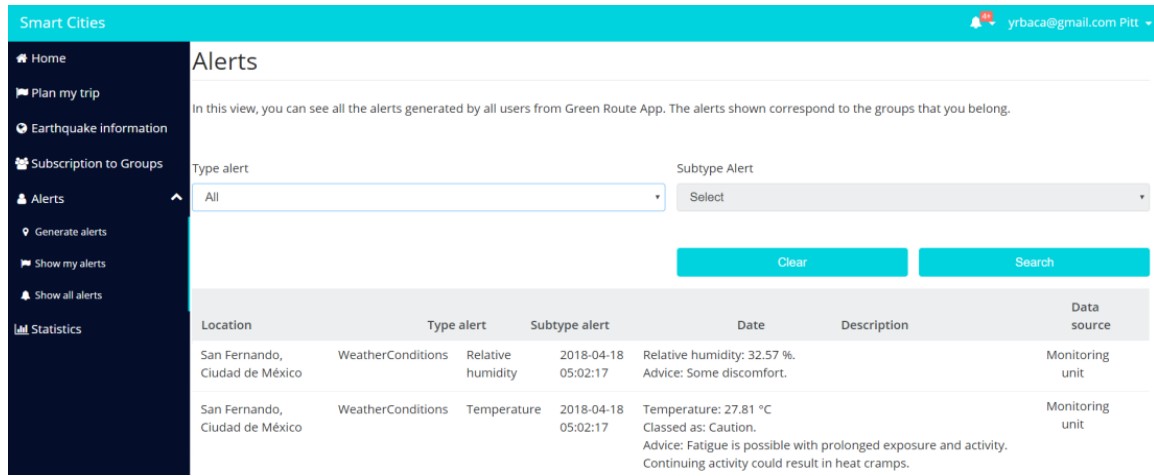

**Figure 8.** Data obtained from the Air Quality Monitoring Unit published in Green Route.

The data collected by the Air Quality Monitoring Unit is also shown in a map in the Green Route Application (see Figure 9).

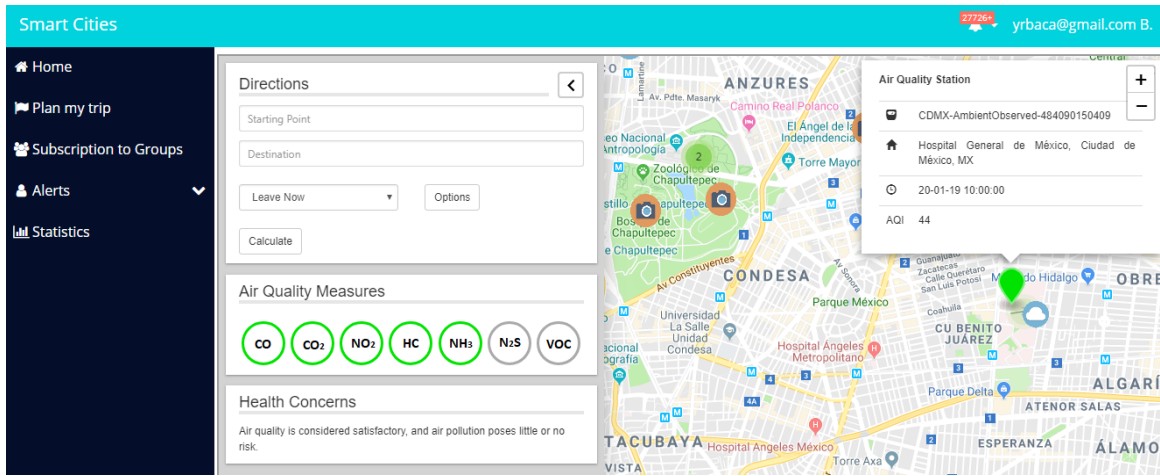

**Figure 9.** Air Quality Monitoring Unit data on a map.

## 6. Conclusions

The IoT paradigm pushes modern sensor networks to monitor a wide range of phenomena in areas such as environmental monitoring, healthcare, industrial processes, and smart cities [25]. In this context, it is important to generate applications with the purpose of measuring the air quality. Many people could benefit by knowing the air quality and trying to avoid the places with more pollution. In addition, this kind of applications can provide enough information to generate statistics and understand of the way pollutants affect the environment. At present, the data sent to the FIWARE Cloud are used for the Green Route mobility application to find routes in a city that are mindful of the pollution and weather conditions, with the aim of improving the life quality of citizens and fostering environmentally friendly behaviors by citizens [23,24].

Due to the increasing pollution in recent years, it is essential to monitor pollutants values to prevent the consequences of an excessive concentration, reduce the pollution production, and avoid the contact with major pollutant concentration through IoT tools. This kind of IoT tool for the monitoring and sharing of the data in an effective system allows us to manage the information in a smart way in order to improve the knowledge of the problem and, consequently, take preventative measures in favor of the urban air quality and human health [26].

Additionally, this kind of application allows people to identify if they are exposed to polluted air, and this will probably have an impact on their mobility behavior and on their awareness of air pollution. It could be assumed that people will begin to explore the air quality of different areas of the city. As a result of a greater awareness of air pollution, people could start to question their own mobility habits, propitiating a positive impact on air quality.

Finally, one of the main components of the Air Quality Monitoring Unit is the Cloudino, which was designed thinking in three main characteristics to take to reality the vision of the IoT: Small size, easy use, and low-cost hardware. With these characteristics, the Cloudino allows to everyone the possibility to incorporate IoT technologies in their projects without any technical or economical limitations.

**Author Contributions:** Conceptualization, Y.R.B.G. and A.M.R.; Data curation, Y.R.B.G.; Methodology, Y.R.B.G. and H.E.E.; Software, Y.R.B.G.; Validation, Y.R.B.G., H.E.E., and A.M.R.; Investigation, Y.R.B.G.; Resources, H.E.E.; Writing—original draft preparation, Y.R.B.G.; Writing—original draft, Y.R.B.G; Writing—review and editing, A.M.R., Y.R.B.G., and D.V.V.; Visualization, Y.R.B.G. and D.V.V.; Supervision, Y.R.B.G. and A.M.R.; Project administration, Y.R.B.G. and H.E.E.; Funding acquisition, H.E.E.

**Funding:** This research was funded by CONACyT, grant number FONCICYT 24/XI-E/2016 with the project called *A FIWARE-based SDK for developing smart applications—SmartSDK*.

**Conflicts of Interest:** The authors declare no conflict of interest.

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
