# Peer review of "A Novel Air Quality Monitoring Unit Using Cloudino and FIWARE Technologies"

_mca, doi:10.3390/mca24010015_

Reviewer 1 Report

The current work is aimed to present an air quality IoT node to monitor cities using Cloudino and FIWARE Technologies. There are some improvements so that the work can be published. Next, they are detailed:

A paragraph with the main contributions is needed in the introduction.

Section Pollutants Measured and its Effects to Human Health doesn't add novelty

The work doesn't detail related works, please could you add a new section or try to justify why is missing

The main contribution is not clear. I think that the green route application should be described in more detail

The use case (cities) should be presented

A pdf with some notes is enclosed

Author Response

A paragraph with the main contributions is needed in the introduction.

A paragraph with the main contributions has been added in the Introduction.

Section Pollutants Measured and its Effects to Human Health doesn't add novelty

The research takes into account almost all of the most common pollutants for the Air Quality Index. This information is specified in a section called Background about Air Quality pollutants

The work doesn't detail related works, please could you add a new section or try to justify why is missing

The related work section has been added.

The main contribution is not clear. I think that the green route application should be described in more detail

The Green Route application is described in more detail in the section called Data consumption by users and applications

The use case (cities) should be presented

The is collected from Mexico City, and it is mentioned in the Introduction and in Results and Discussion section.

A pdf with some notes is enclosed

The references suggested were added.

Reviewer 2 Report

This research entitled "A Novel Air Quality Monitoring Unit Using Cloudino and FIWARE Technologies " is very interesting and is suitable to accept to be published in your Journal. However, please check that 

1. What kinds of sensors are employed in this research?

2. How are the measured vales of each sensor shown on GUI of OCB?

Because these two points should be conducted obviously so as to be convenient and be easy to be used by users in real situations.

Author Response

1. What kinds of sensors are employed in this research?

The sensors are described in the section 4.1 Sensors and electronic components.

2. How are the measured vales of each sensor shown on GUI of OCB?

There is no GUI of OCB. The OCB is a REST API, and the data can be consumed by users or applications only trough REST calls. In this case, the data is consumed by the Green Route application, and in this application a user can see the measured values. There are also some Tools that facilitate the execution of REST calls, such as Postman or Insomnia. In the section 4.4.1 Orion Context Broker, it is mentioned that this component is REST API.

Round  2

Reviewer 1 Report

For me, this version looks more rigorous.

Congratulations on completing a high quality paper